# Low-Cost Self-Calibration Data Glove Based on Space-Division Multiplexed Flexible Optical Fiber Sensor

**DOI:** 10.3390/polym14193935

**Published:** 2022-09-20

**Authors:** Hui Yu, Daifu Zheng, Yun Liu, Shimeng Chen, Xiaona Wang, Wei Peng

**Affiliations:** 1School of Physics, Dalian University of Technology, Dalian 116024, China; 2School of Optoelectronic Engineering and Instrumentation Science, Dalian University of Technology, Dalian 116024, China; 3Marine Engineering College, Dalian Maritime University, Dalian 116026, China

**Keywords:** optical fiber, data glove, wearable devices, flexible sensor, human-robot interaction

## Abstract

Wearable devices such as data gloves have experienced tremendous growth over the past two decades. It is vital to develop flexible sensors with fast response, high sensitivity and high stability for intelligent data gloves. Therefore, a tractable low-cost flexible data glove with self-calibration function based on a space-division multiplexed flexible optical fiber sensor is proposed. A simple, stable and economical method was used to fabricate flexible silicone rubber fiber for a stretchable double-layered coaxial cylinder. The test results show that the fiber is not sensitive to the temperature range of (20~50 °C) and exhibits excellent flexibility and high stability under tensile, bending and torsional deformation. In addition, the signal detection part of the data glove enables compact and efficient real-time information acquisition and processing. Combined with a self-calibration function that can improve the accuracy of data acquisition, the data glove can be self-adaptive according to different hand sizes and bending habits. In a gesture capture test, it can accurately recognize and capture each gesture, and guide the manipulator to make the same action. The low-cost, fast-responding and structurally robust data glove has potential applications in areas such as sign language recognition, telemedicine and human–robot interaction.

## 1. Introduction

Wearable devices based on flexible electronics have experienced tremendous growth and development over the past two decades [1,2,3]. Among them, the data glove is one of the most popular wearable devices. With the popularity of virtual reality (VR) applications, data gloves have been widely used. The data glove is a multi-sensor device that generates large amounts of data. It is more complex than other input devices [4]. However, most researchers still use the data glove because its natural interaction with humans is a way to improve system operation and is suitable for many specific fields. At present, data gloves have been increasingly used in the fields of remote operation and robot control [5,6], surgical training for medical applications [7,8], entertainment and sports for VR systems [9] and sign language recognition [10]. 

A person wearing a data glove can handle hazardous materials or machinery by remotely controlling a manipulator or gripper in a hazardous environment (such as outer space) or in a factory [11,12]. In the medical field, wearable strain-sensing devices allow surgeons to perform surgery remotely, due to their extreme accuracy and usability. Similarly, they can allow students to practise surgery in a simulated environment, reducing the need for cadavers or living organisms [13]. Data gloves and wearable devices have been widely used in virtual reality, especially in the rapid development of the video game industry [14]. The strain sensor-based data glove accurately captures gestures and orientation during the creation of virtual images to capture complex hand movements in a comprehensive way. Data gloves have been used to identify sign language communication between deaf people and those who do not understand sign language [15,16]. Deaf people wear gloves when communicating through sign language, which is then translated into spoken or written language for the recipient to use. This allows deaf people to communicate effectively with the entire population and has been successful enough for them to do so in several languages. However, wearable devices such as data gloves still need to overcome many difficulties before being widely promoted and popularized in daily life. For example, the lack of tractable sensors with excellent sensing performance, the need for bulky and expensive external equipment, and the lack of real-time output of intuitive signals may result in poor signal quality and a user-unfriendly experience, thus limiting its further applications [17,18]. In addition, for flexible design of sensors, it is a challenging task to select the appropriate soft base material to achieve the functions of stretchability, flexibility and torsionability. Silicone rubber, hydrogels and other polymer substrates have previously been selected and can provide a tensile range of 0 to 300% or even greater, which provides a guarantee for the stretchability and reproducibility of strain sensors [19,20,21,22]. In recent years, fiber-based materials have been preferred to make flexible sensors in wearable devices [23]. Compared to block or film compounds, fiber-based materials can be designed in a variety of shapes and mounted on wearable fabrics, meeting the excellent flexibility, breathability and comfort requirements of wearable electronic devices [24,25,26]. However, the process of making fiber is often accompanied by high cost, is time-consuming and environmentally unfriendly, and has other uncontrollable problems [27,28]. Therefore, low-cost and tractable flexible sensors with fast response (response time within 500 ms), high sensitivity, good linearity and super-stability are very necessary for the construction of user-friendly, intelligent wearable devices [29,30].

In this work, we proposed a tractable low-cost flexible data glove with simple sensing mechanism and self-calibration function based on a space-division multiplexed flexible optical fiber sensor. The structure diagram of the data glove is shown in Figure 1a. We adopted a simple, stable, economical and controllable method to fabricate the flexible optical fiber with adjustable and stretchable properties. The flexible optical fiber used for gesture-sensing monitoring has excellent flexibility, stability and accuracy. It not only maintains good sensing performance in the temperature range of 20 °C to 50 °C, but also shows high optical transmission stability after tensile, bending and torsional deformation. In addition, the detection part based on a camera and algorithm can achieve compact and efficient real-time information acquisition and processing. A simple and efficient space-division multiplexed fiber-sensing scheme was adopted to integrate five flexible fibers with the camera and fix them on the glove. This can eliminate the crosstalk signals between optical fiber sensors and save the space and weight occupied by a signal demodulation part. The sensor can be easily fixed to the surface of a normal textile glove, so the size of the data glove can be adjusted according to different types of textile glove. Furthermore, combined with the self-calibration function that can improve the accuracy of data acquisition, the data glove can be self-adaptive according to different hand sizes and bending habits. This enables a “user-friendly” data glove to be customized to the user’s needs, which is a key issue for wearable devices. In consequence, the flexible and customizable data glove involves simple fabrication techniques and common sensor components, achieving real-time monitoring of joint movements and gesture capture at a relatively low cost (the actual price of each component is recorded in Appendix A and the price comparison of data gloves of different brands and models is recorded in Appendix A). This will help to significantly expand the applications of data gloves in motion monitoring, telemedicine, human–robot interaction and VR.

## 2. Materials and Methods

### 2.1. Materials and Instrumentation

Double additive liquid silicone rubber (Dongguan Obasiaseal New Materials Co., Ltd., Dongguan, China), double component room temperature vulcanized (RTV) silicone (HASUNCAST Polymer Technologies., Montgomeryville, PA, USA), demoulding agent (Guangzhou Botny Chemical Co., Ltd., Guangzhou, China), acrylic hollow tube (Qingdao SNDME Co., Ltd., Qingdao, China), heat shrinkable tube (Shengpai Insulation Material Co., Ltd., Dongguan, China) and a number of positioning plugs (Qingdao SNDME Co., Ltd., Qingdao, China), vacuum pressure pump (Yancheng ricky laboratory Instrument Co., Ltd., (Yancheng, China) 10 L/min) and electric blast drying oven (Tianjin Hongnuo instrument Co., Ltd., (Tianjin, China) 101-00BS). The stepper motor (Beijing Times Chaoqun Electrical Technology Co., Ltd., (Beijing, China) CBX1605-150) was used to generate tensile strain for the flexible optical fiber. The sliding table (Taihe Qicheng Hardware & Building Materials Co., Ltd., (Beijing, China) RS60-L) was used to apply bending and torsional strains on the flexible optical fiber. 

### 2.2. Fabrication of Flexible Optical Fiber

As shown in Figure 1b, when the optical fiber is stretched or bent the propagation path of light in the optical fiber changes cannot satisfy the critical angle requirement of total internal reflection. The propagation mode of light in the optical fiber changes from transmission mode to radiation mode, so that part of the light energy penetrates into the cladding or leaks outward through the cladding, resulting in loss. However, because the hand has soft tissues such as muscle and skin, the movement of one knuckle must cause the adjacent knuckles to move together. In this case, the measurement of optical loss becomes very complicated, which leads to the increase of experimental errors and the decrease of the reproducibility of experimental conclusions. Therefore, the curvature of the finger is determined by measuring the total loss of the entire finger. This is the basis of the data glove’s ability to capture gestures. Figure 1c shows the linkage between data glove and manipulator.

The flexible optical fiber in this work consists of a double-layered coaxial cylindrical silicone rubber structure. The core and cladding are made of high refractive index silicone rubber (RI = 1.413) and low refractive index silicone rubber (RI = 1.411), respectively. The finished flexible optical fiber is shown in Figure 2a. The cross-section diameters of core and core-cladding structures are 1.5 mm and 2.9 mm, respectively. The length of finished fiber is 25 cm. 

As shown in Figure 2b, the fiber can be stretched, bent and twisted at will due to its certain toughness and recoverability. The Figure 2c shows the manufacturing process of the flexible optical fiber. In order to ensure the fabrication process of the flexible optical fiber was less disturbed by external debris, we adopted the process of making cladding first and then filling the core to the cladding. The first step was to prepare the two-component additive liquid silicone rubber required for cladding and the two-component RTV silica gel required for the fiber core. The material solvent and crosslinking agent were mixed in the two components according to the weight ratio of 10:1. After thorough stirring, the liquid was put into the vacuum pressure pump for defoaming. The liquid after defoaming was injected into the syringe and stood for five minutes until no bubbles were generated in the liquid (bubbles affect the refractive index and toughness of the fiber). In order to make demoulding easier, it was necessary to advance a layer of oily demoulding agent on the mold. When the demoulding agent was dry, the heat shrinkable tube (inner diameter: 1.5 mm) was placed in the acrylic hollow tube (inner diameter: 2.9 mm). The heat shrinkable tube was fixed in the center of the acrylic hollow tube with a positioning plug. The defoamed cladding solution was injected into the interlayer between the acrylic tube and the heat-shrinkable tube with a syringe. This was let stand for 5 min until no bubbles emerged. Then it was placed in the drying oven at 50 °C until the curing process was complete. The drying oven temperature was 90 °C and the heat shrinkable tube heated until it was reduced to 1/3 of its original diameter and then it was pulled out. After it was cooled to room temperature, a stretchable and transparent hollow tube was formed, which was the cladding of the flexible optical fiber. The bubble-free RTV silicone rubber was injected into the hollow tube with a syringe. Tiny bubbles appeared in this process. Therefore, we left it in the natural environment until the bubbles were discharged. After confirming that there were no bubbles in the fiber, it was placed in the drying oven at 95 °C until the curing process was completed. After it was cooled to room temperature, the flexible optical fiber was pulled out of the acrylic tube. 

### 2.3. Integration of the Flexible Fiber Sensors into Data Glove

To accurately quantify finger movements, the flexible fiber should fit well into the glove. If the flexible optical fiber is directly fixed to the glove, the long-term compression of the optical fiber at the knuckle may shorten its service life due to its high curvature. Therefore, indirect fixation was adopted to fix the optical fiber on the gloves. The flexible optical fiber was embedded in the black latex tube. The two ends were then fixed with black silicone rubber. The black latex tube was first treated with plastic surface treatment agent, and then fixed on the gloves with soft glue, so as to avoid other unnecessary stimulation of the flexible optical fiber. This increased the deformation area of the flexible fiber and dispersed the pressure at the knuckle. 

The flexible fiber sensor quantifies the optical fiber deformation by measuring the total optical loss of the whole finger during propagation. Therefore, it is important to choose the right light source. The common light source is large in size, which makes it difficult to install on gloves and gives it limited flexibility. The selection of surface- mounted device (SMD) red beads with an Incidental line LED patch (2.0 mm × 1.2 mm) is a good solution to this problem. We fixed the beads in a round plug (outside diameter: 5 mm). The wire was built into the glove. Two button batteries provided power for the five beads. The other end of the round plug was suitable for the insertion of the flexible fiber to realize the direct coupling between the light source and the flexible fiber. A thin silicone rubber sheet was fixed on the back of the glove. The silicone rubber sheet was a flexible substrate, which not only ensured the softness and flexibility of the back of the hand, but also ensured the constant relative position of the flexible optical fiber during stretching. 

In order to carry out simultaneous, reliable and real-time monitoring upon the changes of fiber-optic signals corresponding to the five fingers, we adopted a simple and efficient space-division multiplexed fiber-optic sensing scheme. This can not only eliminate the crosstalk signal between optical fiber sensors, but also save space and weight occupied by the signal demodulation part. As shown in Figure 1a, five flexible optical fibers are respectively integrated with the camera and fixed on the glove, so that the camera can collect the gray values for the cross-sections of the five flexible optical fibers. The fiber was fixed with epoxy filler according to its thickness, so that the cross-section of the fiber could directly face the camera lens. A layer of silica gel was applied to the molded model to ensure that the flexible fiber could not move during finger movement. We used an 8-megapixel mini camera (38 mm × 38 mm) with a dynamic frame rate of 25 fps~30 fps. In order to make the integrated module more portable and easier to install on the glove, a wide-angle lens without distortion was configured for the camera, which enabled the camera to completely collect all gray value information in a limited range. When the finger was bent, the brightness of the flexible fiber on the finger would change rapidly. The bending degree of the finger could be judged by capturing the gray value from the end face of each flexible fiber using the camera. The acquisition module was encapsulated with a black matted acrylic sheet (2 mm) to prevent the information collection process from being interfered with by external light. In addition to acrylic adhesive for fixing the contact surface, epoxy putty was also used to fill and fix the interface between the camera and the data glove, so that there was no relative movement between the data glove and the camera. 

To sum up, the data glove was based on all-fiber sensing, which could shield electromagnetic interference. The fiber was not combined with any other materials (such as Au, Ag or GO), and the main structure was stable (no reorganization, such as polishing and splicing). Moreover, the sensor could be conveniently fixed to the surface of a normal textile glove. So the size of the data glove can be adjusted according to the type of textile glove, which could improve the applicability of the data glove.

## 3. Results and Discussion

### 3.1. Performance Characterization of Flexible Fiber

In order to quantitatively evaluate the mechanical and optical properties of flexible optical fibers, a series of performance tests were carried out, including tensile strain, bending and torsion tests. The strain-normalized intensity curve of core-clad fiber at wavelength 650 nm is shown in Figure 3a. It can be seen that the optical fiber retains its mechanical integrity under 90% strain. When the strain increases from 90% to 100%, the strain-normalized intensity curve of the fiber shows a breaking point: A. This indicates that the fiber will not completely break until the strain is greater than 90%. In addition, based on the linear region of the strain-normalized light intensity curve, the tensile strains ranging from 0% to 50% were applied to the fiber at a strain interval of 10%. The normalized spectra under different strain states were measured. These are shown in Figure 3b. With the increase of the tensile strain applied to the fiber, the normalized light intensity through the fiber gradually decreased. When the elongation of the fiber reached 50%, the intensity of light passing through the fiber was lost by about 10%. The inset in Figure 3b shows the strain-normalized intensity curve of the fiber at the wavelength of 650 nm during stretching. It shows that the normalized light intensity of the fiber decreases by about 9% as the tensile strain of the fiber increases from 0% to 50%. This is basically consistent with the curve in Figure 3a. Therefore, the deformation sensor selected the SMD red beads with LED patch as the light source. 

To further study the mechanical properties of flexible optical fibers, we defined a period of stretching strain from 0% to 50% and then back to 0%. The fiber was repeatedly stretched for 50 cycles. The normalized spectra of the fiber in the wavelength range of 530~685 nm after each cycle are shown in Figure 3c. In this cycling process, the normalized light intensity in the fiber decreased from 1.00 to about 0.98 after the first cycle. But in the following 49 cycles, the normalized light intensity in the fiber gradually stabilized between 0.94 and 0.98 after each cycle. This is due to the Mullins effect, which describes the stress softening and hysteresis that occurs in elastic materials [31]. Therefore, to ameliorate the adverse effects, each fiber used for sensing was pretreated at least 50 times in a cycle from 0% strain to 50% strain. In practice, the pretreatment was always performed at a higher strain than the maximum strain encountered by the sensor in use. In order to characterize the optical properties of the fiber, we quantified its propagation loss by successively shortening the fiber and measuring its normalized light intensity. In the wavelength range of 530~685 nm, the normalized intensity increased with the decrease of fiber length (Figure 3d). 

After evaluating the mechanical and optical properties of the fiber separately, we discussed in detail the wavelength-dependent optical response of the fiber to disturbances caused by repeated stretching, bending, and torsion. For the purpose of quantifying the effect of stretching-on-optical-transmission characteristics of the fiber, the fiber was cyclically stretched for 50 times at 20 °C, 30 °C and 50 °C within the strain range of 0% to 50%. And the normalized spectra of the fiber after each cycle were measured (Figure 3e–g). The cycling process at 20 °C is shown in Figure 3e. The normalized light intensity in the fiber decreased from 1.00 to 0.98 after the first cycle due to the Mullins effect. However, during the second to 50th cycles, the normalized intensity in the fiber was distributed between 0.93 and 0.98. In other words, the normalized intensity was lost by up to 5%. Figure 3e shows the cycling process at 30 °C. Similarly, after the first cycle, the normalized light intensity in the fiber dropped from 1.00 to about 0.97. In comparison, in the remaining 49 cycles, the normalized intensity in the fiber was between 0.90 and 0.97, meaning that the normalized intensity lost up to 7%. Figure 3g shows the cycling process at 50 °C, which is similar to the cycle at the above two temperatures. After the first cycle, the normalized light intensity in the fiber decreased from 1.00 to about 0.96. But during the second to 50th cycles, the normalized intensity in the fiber was distributed between 0.87 and 0.96, and the normalized intensity lost up to 9%. By comparison (Figure 3e–g), it can be found that the light loss gradually increases with the rise of temperature after the cycle. However, when the ambient temperature is below 50 °C, the loss of normalized light intensity is less than 10%. The above results have demonstrated flexibly that the fiber can maintain high stability when operating below 50 °C. 

In addition, the strain-normalized light intensity curves of the fiber at the wavelength of 650 nm during the first and 50th cycles at room temperature were measured, as shown in Figure 3h. By comparing the normalized light intensity corresponding to each strain during the two cycles, the light loss caused by stretching strain is only about 5% after 50 cycles of stretching. Fibers exhibit a reduction in normalized light intensity as they elongate, due to wavelength dependence. This can be reversible and repeatable for at least 50 cycles. The inset in Figure 3h shows that the light transmitted in the fiber decreases with the increase of the fiber length due to the longer path of the light. This effect is also observed in other stretchable fibers. While tensile strain occurs, the optical signal in the flexible fiber will change obviously. This can satisfy the requirements of deformation sensor. 

The fiber is also repeatedly affected by bending and torsion. These two additional stimuli are important in sensing applications. For the bending response evaluation, the bending angle *θ* is defined as shown in Figure 4a. The fiber was bent from 0° to 120° and then back to 0° as one of 100 cycles. Normalized spectra of the fiber were measured after each cycle (Figure 4b). Similar to the tensile experiment, the normalized light intensity decreased from 1.00 to 0.98 after the first cycle in the cyclic bending process. Nevertheless, the normalized light intensity in the fiber was distributed between 0.96 and 0.98 during the second cycle to the 100th cycle, namely, the loss is about 2%. In the torsion test, the optical fiber was twisted from 0° to 90° and then restored to 0°, as one of 100 cycles. And the normalized spectrum of the optical fiber after each cycle was measured (Figure 4d). There was no phenomenon similar to the Mullins effect during the torsional cycles. After 100 torsional cycles, the normalized light intensity in the fiber was distributed between 0.98 and 1.00, meaning that the loss of normalized light intensity is lower than 2%. The transmission loss observed in the above experiments is due to the fact that the light propagating in the fiber cannot satisfy the critical angle requirement of total internal reflection. 

In the bending test, the angle-normalized intensity curves at 10 intervals during the 10th to 100th cycles and the 1st cycle were measured (Figure 4c). The curves of the 11 cycles basically coincide. In addition, the optical signal varies with the bending angle in the bending process, which satisfied the requirements of the deformation sensor. Similar to the bending experiment, the angle-normalized intensity curves during the 1st, 50th and 100th cycles in the torsion test were obtained as shown in Figure 4e. The light intensity difference at each torsion angle is tiny during the three cycles. The inset in Figure 4e shows that while the torsion angle is increased to 90°, the optical signal in the fiber will change. In fact, when the sensor is deformed, the torsion angle of the fiber is not more than 30°. So, the loss caused by the torsion will not have a significant impact on the measurement of the optical signal. 

In conclusion, the change of ambient temperature and the deformation of stretching, bending and torsion will lead to certain optical transmission losses. When the temperature is below 50 °C, the loss caused by tensile deformation is little and the optical fiber can maintain its sensing performance well. In addition, the flexible fiber also shows high optical transmission stability after bending and torsional deformation. These tests indicate that our optical fiber can not only maintain high stability after deformation, but also that the optical signal in the optical fiber varies with shape variables during deformation, so as to satisfy the requirements of the deformation sensor suitable for data gloves. 

### 3.2. Data Collection and Processing

Based on the design of the data glove and the data acquisition method, a manipulator control platform associated with the data glove was built. Here, a Raspberry Pi (RPi) manipulator with 6 degrees of freedom was used. In the RPi environment, Python was used to process the gray value information transmitted by data gloves. And the processing results could be transmitted to the manipulator in real-time to realize real-time gesture capture. As shown in Figure 5, the data collection and processing process includes pretreatment—automatic calibration and data processing. 

#### 3.2.1. Pretreatment—Automatic Calibration

In the process of using data gloves, the initial states may be inconsistent each time the gloves are worn. In addition, the hand shape of each person and the initial value of the light source may be different. Therefore, when using data gloves, it is necessary to adjust the sensor’s zero position and amplitude. In the adjustment process, the user makes a specific gesture in advance and gets a value matching the user’s hand, so as to ensure the performance stability of the data glove.

After calibration, the gray values of the maximum and minimum bending and stretching of each finger at the limit position and the gray values in the semi-grip state can be collected. The gray value at the maximum and minimum bending limits can be determined to the change range of the gray value. The gray value in the semi-grip state corresponds to the location point of the manipulator, so as to obtain a gentler mapping relation (steering gear value and gray value), which is the basis for data processing. As a consequence, combined with the self-calibration function that can improve the accuracy of data acquisition, the data glove can be self-adaptive according to different hand sizes and bending habits. This enables a “user-friendly” data glove to be customized to the user’s needs. 

#### 3.2.2. Data Processing

During data acquisition and processing of the data glove, the real-time performance and accuracy of the control system are the key factors to measure system performance. In the process of gesture-following, the system needs to respond in time and accurately transmit data to the manipulator, so a reasonable and efficient algorithm is particularly important.

The data processing system can be divided into two parts, including pretreatment and data processing modules. The mapping relation between the gray value and a specific gesture can be transmitted to the system for automatic calibration. Then, a new mapping relation is constructed by associating the pre-processing mapping relation with the steering gear value of the manipulator. After pretreatment, the data glove outputs five gray values (corresponding to five fingers) in real-time. And the gray values are transferred to the manipulator through the mapping relation in the system, so that gesture-following can be realized. That is, every time the user of the data glove makes a hand action, the manipulator will make the corresponding action in real-time. In the process of glove use, three fine-tunings of mapping relations were set, in order to prevent the deviation of mapping relations caused by the inaccuracy of maximum and minimum limit gray values and amplitude obtained in the pre-processing stage; this avoids impacting on the user experience of the data gloves. 

Since the hand has muscle, skin and other soft tissue, a finger’s action may affect the operation of the adjacent sensor without action, resulting in the manipulator making the wrong action. In view of this phenomenon, we tested gesture capture several times and found that it followed a pattern. For example, when the middle finger is bent, the corresponding sensor will respond even if the ring finger and index finger do not move. Because such error fluctuations can be “predicted” in advance of testing, intelligent prediction was built into the software to correct them. After a large number of tests and data analysis, various similar patterns were collected to form an action–correlation–prediction library. Compensation function relations can be obtained according to these error patterns to alleviate the misjudgment of fluctuation. 

During data processing, the real-time data response is extremely important in the whole system, so as not to affect the real-time gesture tracking and the user experience of the gloves. In this work, the real-time data response mainly depended on the frame rate of the gray value from the flexible fiber cross-section collected by the camera. The 8-megapixel camera with a dynamic frame rate of 25 fps~30 fps can collect data 18 times per second and contain 90 data points, which is enough to provide a good user experience. 

### 3.3. Dynamic Demonstration of Gesture Capture

In order to accurately analyze the quantitative process of finger bending and intuitively observe the process of gesture capture, six gestures (number 1, number 2, number 3, number 4, as well as the gradual bending of the index finger and the unfolding–clenching–relaxing of the whole palm) were captured based on the data glove’s associated manipulator control system and structure of the manipulator. Figure 6a shows the change in the gray value with the process of the whole-palm-closing action (three continuous states of relaxing–bending–relaxing). It can be seen that the gray values of the five fingers have corresponding changes in the process of the palm from the stretch state to the clenched state. The palm returned to the stretch state; the gray value also recovered at the same time. When the palm is fully expanded, the flexible fiber is in a completely relaxed state without any bending or stretching pressure. So, the brightness of the fiber section is the strongest and the gray value is also the greatest. As the finger bends, the flexible fiber is subjected to both tension and pressure from the knuckle bending inside the black latex tube. The optical fiber deformation causes radiation loss, leading to a reduction in the brightness of the flexible fiber cross-section, and the gray value will also decrease synchronously. When the full palm is clenched, the flexible fiber is stretched and bent to the utmost. At this point, the brightness of the fiber cross-section is at its darkest, and the gray value drops to the base level. After a cycle, the gray value returns to the initial value. In order to evaluate the performance of the flexible fiber sensor more carefully, the index finger was chosen as the representative, its bending process was slowed down, and it was kept for a short time in four obviously different bending states. As shown in Figure 6b, the gray values of fingers in these four bending states have obvious changes. In the process of gradually moving from the finger bending state to the stretching state, with the decrease of the bending angle, the pressure on the flexible fiber decreased, and the gray value increased. In addition, without interference from other fingers, the gray value of the stay state remained unchanged. 

Figure 7a shows the change in the gray value caused by repeatedly making the gesture of number 1 in the expanded state. When the number 1 gesture is made, the gray values of other fingers except the index finger (green curve) change significantly. And the gray values decrease as the fingers bend. (Figure 7b–d), respectively, show the changes of gray values generated by the gestures of numbers 2, 3 and 4 repeatedly from the expanded state. Some similar phenomena can be seen in these three pictures. For example, in Figure 7c, the thumb and index finger are bent when doing number 3. When these two fingers are bending, the curves of the other fingers fluctuate as well, rather than being a straight line as expected. This is due to the physiological structure of the hand. Knuckle movement is not independent. The fingers are linked when making different gestures. However, the correlation fluctuation is not sharp enough to affect the use of the data glove. This experimental result proves that the flexible optical fiber can quantify the finger movement according to changes in the gray value. Obviously, the data glove can quickly recognize and accurately capture gestures by dint of reliable and stable sensing performance. 

To evaluate the timeliness and repeatability of the data glove, it was used to perform several consecutive sets of movements. As shown in Appendix A, the data glove went from the bending of one finger to the gradual display of numbers 1, 2, 3, 4, 5, and then back to numbers 4, 3, 2, 1, to the full grip of the hand. The video screenshots of linkage between data glove and manipulator are shown in Figure 8. The continuous movements of each finger are tracked in real-time by the manipulator, indicating that our data glove can accurately recognize each gesture and respond quickly in the dynamic process, and then guide the manipulator to make the corresponding actions, thus realizing the rapid human–robot interaction. The time delay of the data glove was measured, and the time from the initiation of the action of the data glove to the completion of the following action of the manipulator was about 0.5 s. The time delay can be further reduced by increasing the acquisition frequency of the optical signal to capture a larger number of points. Consequently, the data glove can fully satisfy real-time interactions with a manipulator or VR. There is every reason to expect the data glove to have a great future in artificial intelligence.

## 4. Conclusions

The proposed tractable low-cost flexible data glove with simple sensing mechanism and self-calibration function is based on a space-division multiplexed flexible optical fiber sensor. The flexible silicone rubber fiber for stretchable double-layered coaxial cylinder was prepared by a simple, stable, economical and controllable method. It has the potential for mass production. The fibers show excellent repeatability, stability and mechanical durability in the range of 0~90% tensile strain. In the temperature range of 20 °C to 50 °C, the optical loss is less than 10% when the fiber is stretched repeatedly for 50 times in the range of 0–50% strain. Within the angle range of 0° to 120°, the optical loss generated by the fiber is not more than 4% with 100 cycles of bending. In the angle range of 0° to 90°, the optical loss generated by the fiber is only 2% when the cycle is rotated 100 times. The five sensors integrated with flexible optical fibers and custom data acquisition cameras are mounted on the data glove, respectively. And the detection part based on a camera and algorithm can realize compact and efficient real-time information collection and processing. The sensor can be conveniently fixed to the surface of a normal textile glove, so that the size of the data glove can be adjusted according to the type of textile glove. Furthermore, combined with a self-calibration function that can improve the accuracy of data acquisition, our data glove can be self-adaptive according to different hand sizes and bending habits. In the gesture capture test, the data glove accurately recognized and captured each gesture. The manipulator can respond quickly and make the same action. Our data gloves, with a simple and stable structure, have the characteristics of low cost, easiness to manufacture and customization. The data glove can effectively monitor the movement of finger joints in real-time and do so accurately. The data glove has potential application value in the fields of sign language recognition, telemedicine and human–robot interaction. It is believed that with the development of artificial intelligence (such as PCA, SVM, ANN, and other algorithms will continue to improve), the application of data gloves will be more extensive.

## Figures and Tables

**Figure 1 polymers-14-03935-f001:**
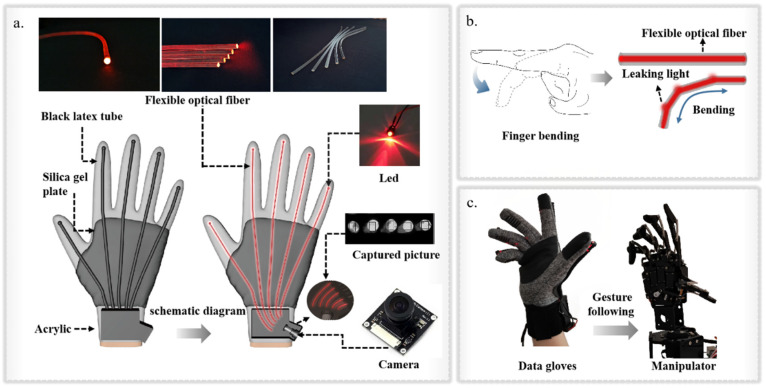
(**a**) Structure diagram of data glove. (**b**) Analogy of flexible optical fiber and index finger behavior during glove bending. (**c**) Linkage between data glove and manipulator.

**Figure 2 polymers-14-03935-f002:**
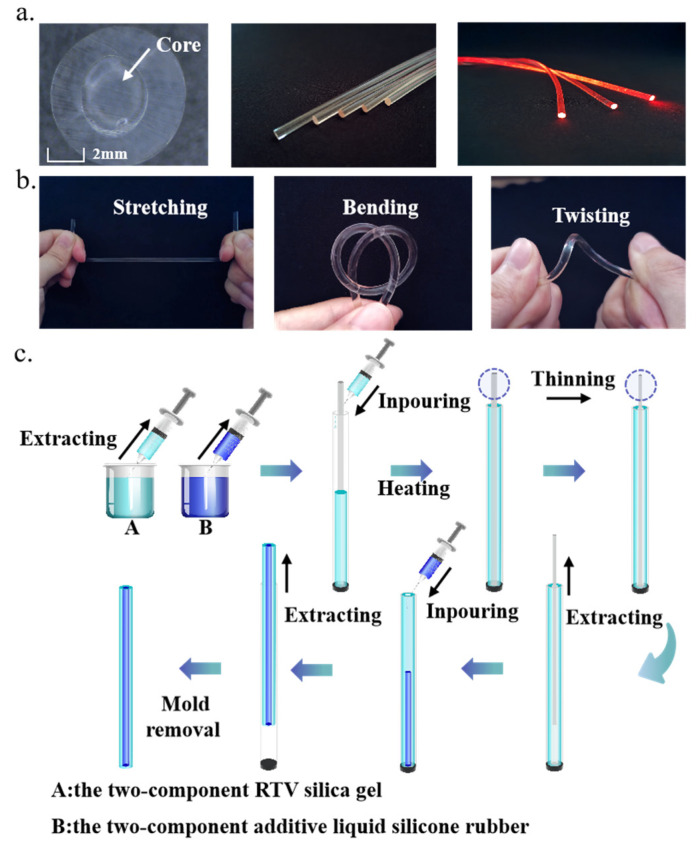
(**a**) The cross-section of flexible optical fiber. (**b**) The state of flexible optical fiber under tension, bending and torsion. (**c**) Manufacturing process of flexible optical fiber.

**Figure 3 polymers-14-03935-f003:**
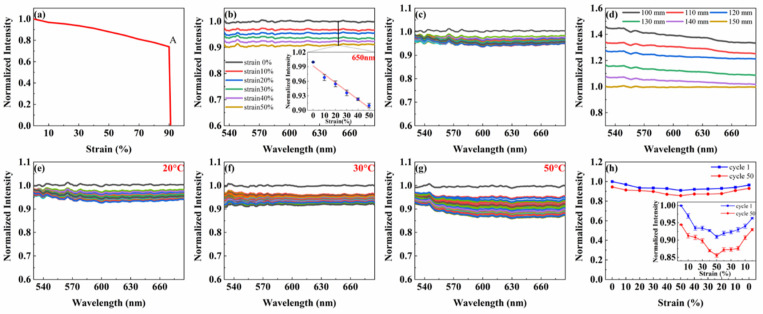
Mechanical and optical properties of the fibers. (**a**) Strain-normalized intensity curve of flexible fiber at wavelength 650 nm (the point “A” indicates that the fiber breaks only when the strain is greater than 90%). (**b**) Normalized spectra of flexible optical fiber with increasing strain from 0 to 50%. The inset shows the relationship between the normalized intensity at 650 nm and the applied tensile strain. (**c**) Dynamic tensile test of a flexible fiber being stretched 50 cycles at the strain range 0–50%. (**d**) Propagation loss of flexible fiber, measured in air by cutback method. Fibers for deformation sensing. Dynamic tensile test of flexible fiber being stretched 50 cycles with 50% applied peak strain at different temperatures, (**e**) 20 °C, (**f**) 30 °C, (**g**) 50 °C. (**h**) Comparison of the first cycle and the 50th cycle in tensile test at room temperature. The different color lines in (**c**,**e**–**g**) represent each stretch cycle.

**Figure 4 polymers-14-03935-f004:**
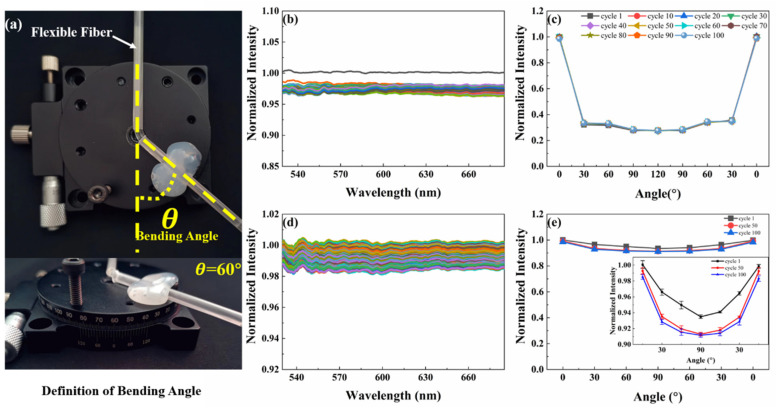
Bending and torsion-sensing performances of the flexible optical fiber. (**a**) Definition of bending angle. (**b**) The flexible fiber is cyclically bent 50 times in the range of 0~120° bending angle. The different color lines represent each bending cycle. (**c**) The angle-normalized intensity curve at 10 intervals during the 10th to 100th cycles and the 1st cycle in the bending test. (**d**) The flexible fiber is repeatedly twisted 50 times in the range of 0~90° torsion angle. The different color lines represent each torsion cycle. (**e**) The angle-normalized intensity curves during the 1st, 50th and 100th cycles in the torsion test.

**Figure 5 polymers-14-03935-f005:**
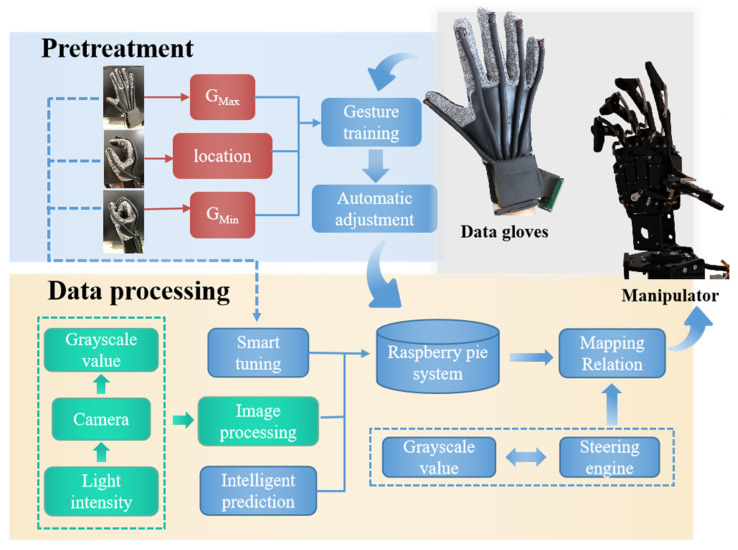
Flow chart of data collection and processing, including pretreatment—automatic calibration and data processing.

**Figure 6 polymers-14-03935-f006:**
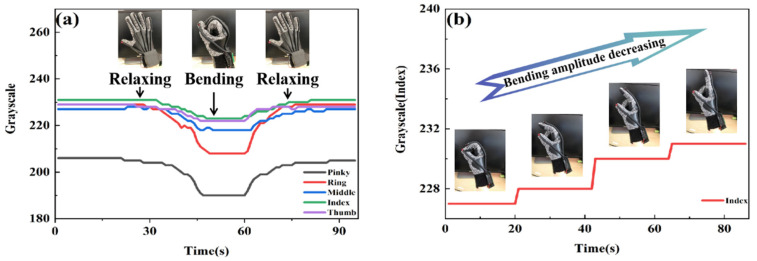
(**a**) Gray value change diagram of the whole palm closing action in three consecutive states of relaxing–bending–relaxing. (**b**) Gray value change diagram of gesture change (four processes of index finger bending).

**Figure 7 polymers-14-03935-f007:**
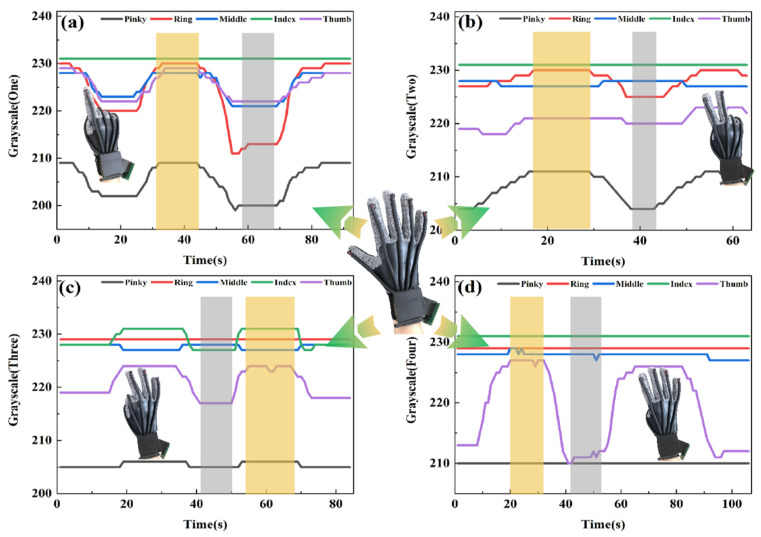
The quantization process of finger bending of data glove and intuitive observation of gesture capture process. The data glove from the stretch state repeatedly makes the (**a**) Number 1, (**b**) Number 2, (**c**) Number 3 and (**d**) Number 4 gestures from the stretch state. The yellow area represents the data block of the five-finger stretch state, and the gray area represents the data block of the corresponding gesture.

**Figure 8 polymers-14-03935-f008:**
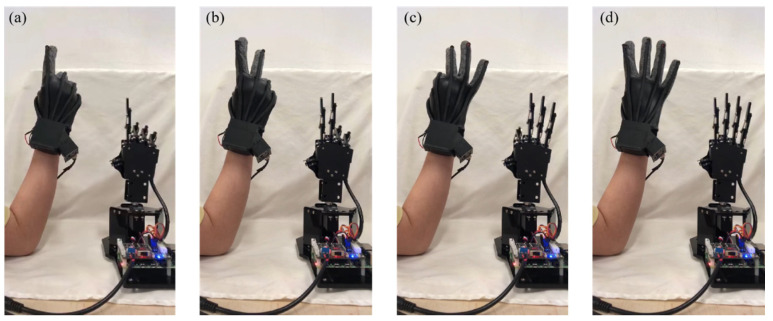
Video screenshots of linkage between data glove and manipulator. The manipulator tracks the hand gestures of the data glove in real-time. (**a**) Number 1. (**b**) Number 2. (**c**) Number 3. (**d**) Number 4.

## Data Availability

Not applicable.

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
