# Peer review of "Low-Cost Self-Calibration Data Glove Based on Space-Division Multiplexed Flexible Optical Fiber Sensor"

_polymers, 2022, doi:10.3390/polym14193935_

Round 1

Reviewer 1 Report

This paper reports a tractable low-cost flexible data glove with a sensing mechanism  very simple and self-calibration function based on SDM flexible optical fiber sensor. So,me comments.

1. first sentence of intro: "Wearable devices based on flexible electronics have experienced tremendous growth and development over the past two decades" - it needs some references in general to consider it and saying flexible electronics and optics. So, please refer: Opto-Electronic Advances, 210098-1-210098-11, 2022; https://doi.org/10.1002/adfm.201902898; https://doi.org/10.1016/j.snb.2021.130794

2.  Fig 2: what the difference compared with the fiber produced in: https://doi.org/10.1364/PRJ.410168  

3. How about the reproducibility of the performance using different probes of this fiber for this results presented? Also, How many cycles of strain for example was performed in each piece of fiber, it means, the repeatability?

4. Error bars analysis is needed in the results presented.

5. when is mentioned in conclusion about the "continuous development of artificial intelligence" to get more conclusions with data collected, I think the authors must mention some algorithms like PCA, SVM, ANN, like explored in some related papers mentioned in the literature  and refer some of them.

Author Response

Please see the attachment. And we have sent the Video R1 to the editor.

Reviewer 2 Report

1-The novelty of this research is not clear and difficult to recognize in the abstract, introduction, and conclusion. 

2-"low cost" is the first word in the title and nothing related to it in the work. the author did not say how it is low cost and where the main part made this design low cost. also, there are no comparison results of cost between the published designs and the author's design.  I suggest a cost schedule ( real prices )  for the author's design and compared it with other designs ( near to real prices ) 

3- literature review in the introduction is weak ( no clear data for this design from other researchers).

4- the fiber sensor design in section 2.3, line 160, how the author can measure the loss of light during propagation? I mean was the author measured the total losses of the full finger or measured the loss at each joint of the finger? 

5- Is it possible to Close up the hand in the author's design?  if possible please include it in the results and discussion part.

6- The life of the fiber sensor design before getting cracked was not estimated

Author Response

(The authors gave the same response as above.)

Round 2

Reviewer 1 Report

The authors need to improve the first point of my comments since the references suggested are not cited all and the authors seems to add more self citations which is not good. Please use the recommended ones.

Reviewer 2 Report

The Author responded positively! 

Author Response

Thank you for reviewing our manuscript and giving us valuable suggestions.